# AGC/AKT Protein Kinase SCH9 Is Critical to Pathogenic Development and Overwintering Survival in *Magnaporthe oryzae*

**DOI:** 10.3390/jof8080810

**Published:** 2022-07-31

**Authors:** Wajjiha Batool, Chang Liu, Xiaoning Fan, Penghui Zhang, Yan Hu, Yi Wei, Shi-Hong Zhang

**Affiliations:** The Key Laboratory for Extreme-Environmental Microbiology, College of Plant Protection, Shenyang Agricultural University, Shenyang 110866, China; wajjiha@syau.edu.cn (W.B.); 2021200150@stu.syau.edu.cn (C.L.); 2021200146@stu.syau.edu.cn (X.F.); zhangph18@mails.jlu.edu.cn (P.Z.); 2021220506@stu.syau.edu.cn (Y.H.); wyziyu@syau.edu.cn (Y.W.)

**Keywords:** MoSch9, *M. oryzae*, overwintering survival, pathogenic development, transcriptomics, conidiophore stalk formation

## Abstract

Primary inoculum that survives overwintering is one of the key factors that determine the outbreak of plant disease. Pathogenic resting structures, such as chlamydospores, are an ideal inoculum for plant disease. Puzzlingly, *Magnaporthe oryzae*, a devastating fungal pathogen responsible for blast disease in rice, hardly form any morphologically changed resting structures, and we hypothesize that *M. oryzae* mainly relies on its physiological alteration to survive overwintering or other harsh environments. However, little progress on research into regulatory genes that facilitate the overwintering of rice blast pathogens has been made so far. Serine threonine protein kinase AGC/AKT, MoSch9, plays an important role in the spore-mediated pathogenesis of *M. oryzae*. Building on this finding, we discovered that in genetic and biological terms, MoSch9 plays a critical role in conidiophore stalk formation, hyphal-mediated pathogenesis, cold stress tolerance, and overwintering survival of *M. oryzae*. We discovered that the formation of conidiophore stalks and disease propagation using spores was severely compromised in the mutant strains, whereas hyphal-mediated pathogenesis and the root infection capability of *M. oryzae* were completely eradicated due to MoSch9 deleted mutants’ inability to form an appressorium-like structure. Most importantly, the functional and transcriptomic study of wild-type and MoSch9 mutant strains showed that MoSch9 plays a regulatory role in cold stress tolerance of *M. oryzae* through the transcription regulation of secondary metabolite synthesis, ATP hydrolyzing, and cell wall integrity proteins during osmotic stress and cold temperatures. From these results, we conclude that MoSch9 is essential for fungal infection-related morphogenesis and overwintering of *M. oryzae*.

## 1. Introduction

Due to global environmental and co-evolutionary changes, emerging plant diseases caused by microbial pathogens are becoming a serious problem and a worldwide concern. Among them, fungal diseases are more prevalent than other pathogen groups [1]. The emergence of new plant diseases due to pathogenic fungi pose a serious threat to global food security, compelling researchers to understand the driving factors responsible for such disease outcomes. One of the key factors influencing fungal disease outbreaks is the pathogen’s ability to produce long-lived spores or resting structures [2]. These are not only the primary inocula in the fungal infection cycle during host-pathogen interaction, but also play an important biological role in terms of enduring viability and protecting the fungal genome during harsh environmental conditions [3].

Usually, the interaction between the host plant and pathogen has two stages, i.e., 1. colonizing the uninfected host population during the growth season, and 2. the pathogen’s survival in a host population during off season (overwintering) [4]. Many microbial pathogens use this evolutionarily-developed technique, namely over-seasoning or overwintering, to survive the most unfavorable conditions (i.e., extreme cold or hot temperatures or in dry conditions) during their life cycle in the form of endospores to spread the infection in the following year [5,6,7]. For example, a study on the bacterial pathogen *Xanthomonas axonopodis* causing bacterial spot has reported that the pathogen may overwinter in the soil by associating with roots of non-host plants [8,9]. Similarly, other studies on *Pseudomonas syringae* [10,11] and *Xanthomonas campestris* [12], responsible for bacterial leaf blight and black rot, respectively, have also found that these pathogens overwinter on seed and un-decomposed celery residue from infected plants left in the field.

Fungal pathogens have also been reported to survive in harsh environmental conditions in the form of specialized structures such as chlamydospores, ascospores, or sclerotia on plant debris or within the soil that the fungus later uses as propagules or as a source of inoculum. A thick, protective exterior cell wall (like in spores) or coat (like in sclerotia) is present on the majority of these specialized survival structures [13]. Chlamydospores formation is common in several of the fungi belonging to phylum ascomycetes such as *Candida albicans* [14] and *Fusarium oxysporum* [15], or to Basidiomycetes such as *Mortierellales* [16] and *Panus* [17] species. The pathogens then infect the hosts by producing the primary inoculum when conditions become favorable, resulting in a disease outbreak in the following year [18,19]. For example, multiple *Fusarium* and *Alternaria* species have been reported to overwinter by using weeds as their alternate host in the absence of their primary host [7]. In addition, multiple wheat pathogens—i.e., *Pyrenophora tritici-repentis* [20] responsible for tan spot, *Puccinia striiformis* [6] responsible for stripe rust, and *Gibberella zeae* [21] responsible for fusarium head blight—have been reported to have overwintering potential due to the presence of their survival hot spots and resting structures (urediniospores or ascospores) in the field during off-season [6,22].

Similarly, rice blast pathogen *Magnaporthe oryzae* has been reported to overwinter through its long-term survival on seeds, infected rice residues in the crop fields, or wild grasses, i.e., foxtails [22,23,24]. Interestingly, the rice blast pathogen hardly forms any resting structure, i.e., chlamydospores, for its survival in harsh conditions. Rather, the pathogen undergoes some genetically determined physiological modification during winter or other unfavorable conditions to use conidia (asexual spores) on any infected rice residues or the mycelium within diseased spots tissues as its main overwintering organs to survive on plant debris [22]. Therefore, studying the genetically determined physiological alteration is especially important in *M. oryzae* during overwintering.

Phosphorylation, a primary means of signal transduction, is the most prevalent sort of post-translational modification in eukaryotes [25,26,27]. In fungal pathogens, research on phosphorylation has gained prominence since it regulates several important virulence mechanisms. Key enzymes of the phosphorylation cascade system are the protein kinases that phosphorylate other proteins using a ubiquitous cofactor, ATP [28]. Mitogen-activated protein kinases (MAPKs) are the serine/threonine kinase proteins that play an important role in the regulation of various developmental processes as well as the transmission of diverse extracellular signals [29]. In *M. oryzae*, three different MAPK signaling pathways have been identified, and most of the studies on these pathways have been elaborated on regulating the appressorium development and other invasive structures [30]. However, no study has yet been reported on the involvement of these kinase proteins in protecting the pathogen from extreme environmental conditions. Here, we have identified a serine/threonine AGC/AKT protein kinase in *M. oryzae* (MoSch9) that has been previously reported to regulate pathogenesis and sporulation [31]. In fungi, Sch9 has already been reported to be involved in signaling pathways that affect fungal filamentation, log-phase growth, nutrient sensing, stress response, autophagy, virulence, etc. [31,32,33,34,35]. However, the role of Sch9 gene in *M. oryzae* cold acclimation capability is still unknown.

Therefore, the main objective of this study is to elucidate the role of MoSch9 in *M. oryzae* overwintering tolerance. For this purpose, MoSch9 deletion mutants were generated to study the respective protein role in morphological development and infection capability of *M. oryzae*. Most importantly, a transcriptomic study was carried out to explore the expression regulation of stress-responsive genes in *M. oryzae* both wild type and ΔMoSch9 mutant strains during warm and cold temperature stress to identify the MoSch9 role in the survival capability of *M. oryzae* during off-season.

## 2. Materials and Methods

### 2.1. MoSch9 Identification and Sequence Alignment

Gene and amino acid sequences of all Sch9 genes reported in different fungi along with MoSch9 (MGG_14773) from *M. oryzae* were acquired from the freely available NCBI database (https://www.ncbi.nlm.nih.gov/, accessed on 23 July 2021). For domain prediction, Pfam domain prediction tool (http://pfam.xfam.org/search/sequence, accessed on 17 August 2021) was used. Later, the amino acid sequences were aligned using an online multiple sequence tool (https://www.ebi.ac.uk/Tools/msa/clustalo, accessed on 20 August 2021) to generate the evolutionary tree using MEGA7. The analysis involved 8 amino acid (Appendix A) sequences, and the evolutionary tree was implied by using the Maximum Likelihood system based on the Poisson correction model. All positions containing gaps and missing data were eliminated. In addition, for predicting MoSch9 protein tertiary structure and its subcellular localization, online tools I-TASSER (https://zhanglab.ccmb.med.umich.edu/I-TASSER/, accessed on 2 September 2021) and Softberry (http://www.softberry.com/berry.phtml, accessed on 2 September 2021) were used, respectively.

### 2.2. Fungal Isolates, Growth, and Storage Conditions

*Magnaporthe oryzae* strain Y34 (JJ88), used as a wild type and as a background for generating knockout mutants for the MoSch9 gene, was isolated and purified from the Jijing88 rice cultivar from Jilin Province, China. Competent bacterial cells, prepared using *E. coli* strain DH5α, were used for the multiplication of constructed plasmids and were cultured in a sterilized lysogeny broth (LB, pH: 7.0) medium. For protoplast preparation, RNA isolation, and gDNA extraction, individual strains were cultured in liquid CM and incubated in a rotatory shaker at a speed of 110 rpm at 25 °C for 3–4 days. For normal fungal growth, storage, and conidiation, complete media and rice bran media (RBM pH: 6.0–6.5) were used following the previously described protocol for media preparation, growth, and storage [36]. For conidiogenesis, respective strains were inoculated on RBM plates and were incubated for 7 days at 25 °C in the dark before scratching hyphae. After scratching, plates were placed under a fluorescent light for 3 days [37,38], and then relative assays were performed.

### 2.3. Targeted Gene Deletion of MoSch9 Mutants and Generation of Complementation Strains

Knockout vector pCX62 was used for deleting MoSch9 gene in JJ88 using the split marker approach. First, around 1kb upstream and 1kb downstream regions were amplified using MoSch9 AF/AR and MoSch9 BF/BR primers, respectively (Appendix A), and then ligated with a hph cassette fragment within the pCX62 vector, and then amplified with primers HYG/F+HY/R and YG/R+HYG/R following the same protocol as reported before [39]. For protoplasts formation and transformation, a standard protocol was followed [40] using a homologous recombination technique. Transformants were selected on hygromycin B-supplemented TB3 medium [40]. Prescreening of all the potential transformants using gene-specific ORF and UAH primer pairs was carried out through PCR assays and later confirmed with RT-qPCR primers (Appendix A). MoSch9-comp vector was constructed by amplifying a 3.8kb fragment containing the native promoter region and complete MoSch9 ORF sequence using MoSch9compF/compR primer pair as described [41]. Constructed vectors were transformed into the MoSch9 deletion mutant using JJ88 protoplast for generating the MoSch9 complementation strains, and later transformants were screened with respective ORF primers and qRT-PCR primers [42].

### 2.4. Assays for Conidial Production, Germination, and Appressorium Development

For conidiophore assay, fungal blocks from each strain on 7 day old RBM plate were cut using a blade and placed on micro slides to keep under light at 25 °C. The prepared samples were later observed at 12, 24, and 48 h using a light microscope (Nikon Eclipse 80i). For staining the conidiophore stalks, lactophenol cotton blue was used 2 min before observing under a microscope following the previously described protocol [41]. Additionally, for the conidia count, 7 day old strains on RBM plates were incubated in a 25 °C light incubator for 3 days. Spores were then collected using sterile water in 2 mL Eppendorf tubes and counted under a microscope with a hemocytometer.

For spore germination and appressorium formation assay, collected conidia spores in 2 ml Eppendorf tubes were adjusted to 1×10^5^spores/mL and added to hydrophobic plastic coverslips as droplets. The coverslips were placed for 2–12 h in a dark, humid incubator at 25 °C temperature, and germination and appressorium development was observed under a microscope at 2 h intervals. At least 100 conidia were observed for each strain per experiment. For appressorium-like structure development, fungal blocks of each strain were placed on barley leaves and on hydrophobic slides and then incubated in a dark and humid 25 °C incubator, and examined under an optical microscope at 24, 48, and 72 hpi. All experiments were conducted in triplicate.

### 2.5. Pathogenicity, Rice Root Infection, and Penetration Assays

For the hyphal-mediated infection assay, hyphal blocks of each strain were inoculated on intact and injured leaves of 3 week old rice seedlings (*O. sativa* cv. Lijiangxintuanheigu) and also of 1 week old barley seedlings. The inoculated leaves were placed in a dark, humid chamber for 24 h and then transferred to a 25 °C growth chamber with a photoperiod of a 12 h light/dark cycle. To ascertain MoSch9 spore mediated pathogenicity, the wild type, ΔMosch9-33, and ΔMosch9-41 spores were collected as previously described and were adjusted (5 × 10^4^ conidia/mL in 0.1% gelatin) suspension. A volume of 2 mL of a conidial suspension was then sprayed onto the rice and barley leaves. The inoculated leaves were provided with similar conditions as described above. Disease development and blast lesions were observed, and photographs were taken after 7 days post inoculation (dpi) in hyphae and spore-induced infection assays.

For rice root infection assay, hyphal blocks were plugged on the root of germinated seed soaked for 3 days in water and placed on 2% agar plates. The plates were sealed and incubated in a humid growth chamber. After 5 days, lesions were observed, and photos were taken [43]. For monitoring infection penetration assay, hyphae and spore suspension of each strain were inoculated on the backside of barley leaves with similar conditions as of pathogenicity assay. Fungal colonization was monitored at 24, 48, and 72 hpi under a light microscope [41].

### 2.6. Stress and Freeze Tolerance Assay

For osmolytes stress assays, the colony growth of the 7 day old strain on CM plates was augmented with various cell membrane stress-inducing chemicals, i.e., 0.1 and 0.2 mg/mL CR (Congo Red), 0.005% and 0.01% SDS (sodium dodecyl sulfate); osmotic stress-inducing chemicals NaCl (5% and 7.5%) and sorbitol (5% and 8%), and hydrogen peroxide (7 mM, 10 mM) for inducing oxidative stress. Their growth was measured to investigate MoSch9 deletion mutant’s stress response. Each test was repeated at least three times. For temperature stress assay, the growth diameter of each strain was measured after 10 days on RBM plates kept in incubators with different temperatures, i.e., 25 °C (control), 33 °C (high temperature stress), and 10 °C (cold stress). Each experiment was repeated at least three times with 6 replicates of each strain for each stress at a time.

For hyphal freeze tolerance assay, respective mutants and wild-type strains were grown on CM plates at 25 °C for 7 days, and then punched with a 9 mm cut to yield multiple fungal blocks for each strain, and finally incubated in a 4 °C refrigerator for 1 day before subjecting to −20 °C treatment. Later, 3 blocks of each stain kept at −20 °C were reactivated on CM at 25 °C every two days, and the number of days until the respective strains could no longer be revived was noted [44]. For studying conidial freeze tolerance, respective mutant and wild-type strains were grown on RBM plates at 25 °C. After 7 days, the plates were scratched and kept under light for 2 days to initiate sporulation. After 2 days, the plates were kept at 4 °C for 1 day, and then spores were collected by scratching the plate’s surface using ddH2O. Collected ΔMosch9 and wild-type spore suspensions in 1.5 mL Eppendorf tubes were placed at −20 °C for 8, 16, 24, and 48 h. The suspensions, after thawing on the ice, were plated on 1.5% Rice extract to ascertain the % of viable spores [45].

### 2.7. RNA Extraction, Real-Time qPCR, and RNA Seq Analysis

For RNA extraction, wild-type JJ88 (Y34), ΔMosch9-33, and ΔMosch9-41 strains were cultured in liquid CM for 4 days, as described above. Total RNA from each strain was extracted using the TRIzol reagent kit (Invitrogen, Carlsbad, CA, USA), stored in a dark container at 4 °C, following a prescribed method [46]. Standard laboratory precautionary measures were taken to handle the TRIzol reagent. The RNA samples were then stored in a −80 °C refrigerator. Using PrimeScript RT regent Kit with gDNA Eraser (RR047A; Takara, Shiga, Japan), cDNA was synthesized from total RNA. Subsequently, a 10 μL reaction mix was prepared containing 5 μL TB green, 0.2 μL of each 10 μM forward and reverse primers listed in Appendix A, and a 5 μL cDNA template. Providing thermocycler conditions, i.e., 2 min at 94 °C, followed by 40 cycles of 94 °C for 15 s, 60 °C for 20 s, and lastly, the melting curve stage of 94 °C for 15 s, 60 °C for 15 s, and 94 °C for 15 s, quantitative PCR (RT-qPCR) was performed using ABI7500 system (Applied Biosystems, Foster City, CA, United States). The relative mRNA expression was calculated using the formula 2−ΔΔCq (Cq = Cq gene − Cq actin) method. qRT-PCR Primers used in our study are mentioned in Appendix A. Actin gene (MGG_03982) was used for positive control. Consistent values were obtained from three biological replicates for each independent experiment. For RNA sequencing, total RNA from wild-type and ΔMosch9 mutant at different temp 25 °C, 33 °C, and 10 °C was extracted and sent to the company (Novogene) for cDNA libraries construction and Illumina sequencing. Expression abundance and differential expression analysis were performed following the previously described methods [42].

### 2.8. Statistical Analysis

Three technical and biological replicates, respectively, for each strain were used for all the experiments. Graphs were made using GraphPad Prism 7 software. For calculating mean ± SD and performing statistical analysis, a t-test was applied using GraphPad Prism 7 and a Microsoft Excel sheet. Error bars were used to denote the SD. ns, *, **, and *** indicate no significant difference (*p* > 0.05), a significant difference (*p* < 0.05), a highly significant difference (*p* < 0.01), and (*p* < 0.001), respectively.

## 3. Results

### 3.1. Phylogenetic Analysis and Expression Profiling of MoSCH9 in Magnaporthe Oryzae

Protein kinase Sch9 and its diverse functional mechanisms have been reported in various species [31,32,33,34,35,47]. Although its orthologs are well conserved in many fungal species [31,32,33,47], there is no up-to-date knowledge on the role of Sch9 during overwintering in pathogenic fungi. Based on the conserved domain sequence of several Sch9 proteins reported in pathogenic fungi (Appendix A), a putative AGC/AKT serine-threonine protein kinase (MGG_14773) was identified in *M. oryzae* genome, termed as MoSch9, with a 2712 bp ORF region, encoding a 903 aa protein, and was used in this study.

Additionally, phylogenetic analysis and domain sequence homology were performed to relate MoSch9 evolutionary lineage with other functionally characterized fungal Sch9 proteins (Appendix A). The analysis results indicated a close evolutionary pattern between MoSch9 and *F. graminarium* and *B. bassiana* Sch9 protein (Figure 1A) compared to other fungal Sch9 proteins, and a high percentage domain homology of 90.06% and 89.16% with *F. graminarium* and *B. bassiana* Sch9, respectively (Appendix A). The protein structure of MoSch9 (Figure 1B) revealed that it is composed of two key domains: a calcium-dependent C2 domain (316–463) and a conserved serine/threonine kinase STKc_Sck1-like domain (498–837). Sequence alignment and Tertiary (3D) structures predicted with the web-based I-TASSER5 revealed a highly conserved amino acid residue at ATP binding site Lys527 and active site Asp624 of MoSch9 (Figure 1C and Appendix A). These results indicate that catalytic sites of Sch9 are well conserved in the fungal group.

Furthermore, to assess the expression patterns of nuclear-localized predicted MoSch9 (Appendix A) at different stages of *M. oryzae* life cycle, we extracted RNA from 5 day old *M. oryzae* Y34 strain grown on CM and RBM media—during *M. oryzae* stalks development 0–9 h, conidium, germination 2–4 h, appressorium formation and 0–5 days of infection cycle—and performed RT-qPCR assay. Taking mycelial growth on CM as a control, we found that MoSch9 expression was significantly upregulated during the initiation of stalk development and conidiation. However, the expression was gradually downregulated at the later stages of stalk development and conidiation, which indicates that the MoSch9 may be involved in activating the conidiophore stalk development genes and the genes responsible for conidiation (Figure 1D). Similarly, we found that during the first 3 days of the infection, expression of MoSch9 gradually decreased and then significantly upregulated on days 4–5 of infection (Figure 1E) indicating that MoSch9 exerts a significant impact on the infectious development of *M. oryzae* during the necrotrophic phase.

### 3.2. Deletion of MoSch9 Gene Has a Significant Effect on Hyphal Melanization and Conidiogenesis in M. Oryzae

As MoSch9 expression was higher during stalk development and the necrotrophic phase of the infection cycle, we speculated that it might be involved in regulating the primary inoculum production for the next infection cycle. To prove our hypothesis, we deployed a homologous recombination approach for targeted gene deletion of MoSch9 gene using the pCX62 vector. We successfully gained two MoSch9 deleted strains (ΔMosch9-33, ΔMosch9-41) by prescreening all the potential transformants through PCR assays and later through qPCR (Appendix A) using gene-specific primer pairs (Appendix A).

To ascertain the role of MoSch9 in the vegetative development of rice blast fungus, we cultured MoSch9 mutant and wild-type strains, on solid and liquid CM media. After 7 days’ growth, assessment of respective strains cultured on a solid medium showed no significant reduction in growth rate compared to wild type (Figure 2A,B). However, visible grayish black ΔMosch9 mycelia, darker than that of wild type, were observed both on solid CM media and in liquid culture, which indicates enhanced hyphal melanization in mutant strains. Moreover, the observed upregulation of melanin-responsible gene expression (Figure 2C) further confirmed that MoSch9 negatively regulates the hyphal melanization in *M. oryzae*, and the deletion of MoSch9 induces the earlier expression of melanization-responsible genes.

Asexual sporulation (conidiation) is the most common reproductive method for this notorious fungus and plays a crucial role in efficient dissemination of pathogenic fungi during infection cycle and their survival under abiotic stress conditions [48,49]. Here, we investigated the potential role of MoSch9 in the asexual reproduction of rice blast fungus by culturing the wild-type and mutant strains ΔMoSch9-33 and ΔMoSch9-41 on RBM and OMA for 7 days. Later, the hyphal scratched RBM plates were exposed to continuous light to induce sporulation. When assessing the spatial distribution of conidiophore stalks and the number of conidia per stalk, we observed a significant reduction (75–85%) in the number of stalks produced by both mutant strains. Moreover, at 48 hfewer stalks bearing 2–3 conidia were observed (Figure 2D,E). In addition, results obtained from the conidia count assay of the collected spores from each strain showed that compared to the wild type, targeted gene deletion of MoSch9 triggered an approximately 90% reduction in number of spores produced by ΔMoSch9-33 and ΔMoSch9-41 (Figure 2F). In addition, the expression of conidiation-related genes, i.e., COS1, COM1, CON6, CON7, HOX6, HOX7 observed using quantitative real-time PCR (qRT-PCR) analysis showed a significant reduction in their transcript level in MoSch9 deleted strains compared to wild type taken as control (Figure 2G). This indicates that MoSch9 contributed positively to conidiogenesis and deletion of MoSch9 compromised the asexual reproduction in *M. oryzae*.

### 3.3. Deletion of MoSch9 Completely Abolished Hyphal Penetration of M. Oryzae into Host Tissues

After observing the sporulation defect in MoSch9 deleted strains, the virulence efficacy of hyphae and asexual spores produced by ΔMoSch9 was investigated. As previously described [36,50], when 7 day old barley and 21 day old susceptible rice leaves were inoculated with mycelial plugs of wild type and mutant strains, MoSch9 deleted strains (ΔMosch9-33 and ΔMosch9-41) showed no lesion on intact and injured leaves of both barley and rice (Figure 3A,B). This indicates that deletion of MoSch9 completely abolished the hyphal mediated pathogenesis of *M. oryzae*. Furthermore, when inoculated with spore suspension, ΔMoSch9-33 and ΔMoSch9-41 strains showed a surface lesion on intact barley leaves, whereas on injured leaves, the extent of blast lesion was significantly reduced in both mutant strains compared to wild type (Appendix A). However, no blast lesions were observed on rice leaves inoculated with Mosch9 mutant spore suspension (Appendix A), indicating that inactivation of MoSch9 completely abolished mycelial- and severely compromised spore-mediated blast infection.

Hyphal-mediated appressorium-like structures and appressorium of *M. oryzae* are the basic infectious structures responsible for the pathogen’s hyphal and spore-mediated blast infection, respectively, in the host plant [36,51,52]. Therefore, to identify the possible reasons that account for attenuation in the virulence of ΔMoSch9 strains, the formation of hyphal-mediated appressorium-like structures and their penetration capability was assessed by inoculating the mycelia of wild-type and ΔMoSch9 strains on barley leaves and hydrophobic coverslips. When observed, mycelium of ΔMoSch9 strains failed to form appressorium-like structures on either barley leaves or hydrophobic surface (Figure 3C), indicating that deletion of MoSch9 completely abolished the formation of hyphal tip appressorium-like structures: the mutant strains were thus unable to penetrate the host cells, resulting in complete loss of hyphal-mediated pathogenesis (Figure 3D). On observing the complete loss of hyphal-mediated pathogenesis, we speculated that the MoSch9 gene might be involved in *M. oryzae* crop root infection, and deletion of MoSch9 may affect the root infection capability of *M. oryzae*. Therefore, we carried out the root infection assay using mycelium plugs of both wild-type and MoSch9 deleted strains to test our hypothesis, following a standard protocol [53]. At 5 dpi, similar results were obtained: blast lesions developed on rice roots inoculated with wild-type Y34, but no lesions were observed on plant roots inoculated with ΔMoSch9-33 or ΔMoSch9-41 strains (Figure 3E).

Similarly, spores formed by wild-type and ΔMoSch9 strains were inoculated on hydrophobic coverslips to observe their conidial germination and appressorium formation characteristics in vitro. When studying the germination capability of MoSch9 mutant strains, a significant reduction in spore germination rate was observed compared to wild type (Appendix A). Similarly, at 8 h, only 19% of the germinated conidia were able to form appressorium (Appendix A) and were still not completely mature, which indicates that deletion of MoSch9 not only delayed the spore germination but also affected appressorium maturation. Furthermore, histopathological examinations showed that delayed germination and appressorium maturation defect influenced the pathogenic development of MoSch9 mutants in host tissues: MoSch9 mutants were able to form the invasive hyphae after 48hpi but were still not able to penetrate nearby host cells compared to wild type (Appendix A). Later, the infection capability of the mutant strain was restored in MoSch9-complemented strain, ΔMoSch9-comp-1 (Appendix A). These results showed that MoSch9 is likely to play an indispensable role in spore-mediated infection and hyphae-mediated pathogenesis, and the deletion of MoSch9 severely compromised both penetration and colonization capabilities of ΔMosch9, thus affecting the successful development of rice blast disease.

### 3.4. MoSch9 Deletion Affects Multiple Environmental Stress Tolerance of Magnaporthe Oryzae

Sch9 has already been reported to be important for multiple stress tolerance of various pathogenic fungi. To confirm this, we assayed the mycelial growth of ΔMoSch9 mutant, and the wild-type strain cultured on CM plates supplemented with different concentrations of cell wall and cell membrane stress-inducing agents, i.e., 0.1 and 0.2 mg/mL CR (Congo Red), 0.005% and 0.01% SDS (sodium dodecyl sulfate); osmotic stress-inducing chemicals, i.e., NaCl (5% and 7.5%) and sorbitol (5% and 8%); and hydrogen peroxide (7 mM, 10 mM) for inducing oxidative stress. After 7 days, when the colony diameter was measured, ΔMoSch9 strain showed more restricted hyphal growth on all the stressed CM plates compared to the wild type, which indicates that deletion of MoSch9 alters the cell membrane integrity (Figure 4A,B) and increases the rice blast fungal sensitivity towards hyperosmotic (Figure 4C,D) and oxidative stress (Figure 4E,F). Later, the hyphal growth of the mutant strain was restored in MoSch9 complemented strain, ΔMoSch9-comp-1 (Appendix A).

### 3.5. MoSch9 Is Associated with Cold Tolerance and Critical for M. Oryzae Overwintering Survival

Considering the upregulation in the expression pattern of MoSch9 during stalks development and during a necrotrophic phase of *M. oryzae* (Figure 1E) along with the hyphal-mediated pathogenicity loss in MoSch9 mutant strains (Figure 3), and their sensitivity towards osmotic stress (Figure 4C,D), we investigated the growth pattern of the respective wild-type and mutant strains on rice bran media under normal (25 °C), high (33 °C) and low temperature (10 °C) to assess their response to temperature stress and so explore the role of MoSch9 in the survival of *M. oryzae* during overseasoning. At normal growth temperature, no visible difference in the colony diameter of wild type and the mutant strains was observed except for the high melanization of the mycelial culture in the mutant strains. However, a significant reduction in the hyphal growth of the ΔMoSch9 strains was observed at high and low temperatures, particularly at low temperature where the mutant strains were unable to grow normally as compared to wild-type (Figure 5A,B).

Furthermore, we observed the freezing tolerance mechanism [44] of respective strains by placing 45 hyphal blocks of each wild type and ΔMoSch9 strains grown on CM media at −20 °C, then reactivating the hyphal blocks at 28 °C after every 2 days and observed the number of days until the strain could no longer be revived. The mutant strains were able to revive until 17 days of −20 °C treatment. However, the wild type could still revive after 28 days of the treatment. In addition, the growth rate was significantly slower compared to wild type (Figure 5C). Similarly, the survival rate of conidia after −20 °C treatment at different time intervals was also remarkably decreased in the mutant strains compared to wild type which indicates the importance of MoSch9 in cold acclimation capability of *M. oryzae* (Figure 5D).

To further validate our result, we checked the expression pattern of MoSch9 in the wild-type and cold tolerance genes [44], i.e., Putative uncharacterized protein (MGG_03133), Glyceraldehyde-3-phosphate dehydrogenase (MGG_01084), Transaldolase (MGG_02624), Catalase-peroxidase 1 (MGG_04337), and SnodProt1 (MGG_05344) in both wild type and ΔMoSch9 strains during temperature stress using qRT-PCR. As expected, around a 2.5-fold increase in the expression pattern of MoSch9 was observed at low-temperature stress (Figure 5E). In addition, the relative expression pattern of cold- tolerance genes in ΔMoSch9 mutant strains compared to the wild type was reduced when observed at normal temperature, and then further downregulated when the mutant was exposed to low temperature. This was especially true for the Transaldolase and Catalase-peroxidase 1 whose relative change in the expression pattern was significantly downregulated (Figure 5F), indicating the importance of MoSch9 in cold acclimation by regulating other important cold tolerance genes that help in *M. oryzae* survival during overwintering and environmental stresses. All these results indicate a key role of MoSch9 in maintaining the survivability of *M. oryzae* during off-season.

### 3.6. Transcriptomic Profile of M. Oryzae Wild Type and ΔMoSch9 Mutant Strains during Cold Temperature Stress

Due to the dramatic phenotype of ΔMoSch9 during cold stress, we carried out the transcriptome analysis between the wild-type and MoSch9 mutant strains at different temperatures (10 °C, 33 °C, and 25 °C). As a result, differentially expressed genes at high and low-temperature stress were listed and highlighted with MoSch9 loss. We found 2786, 4656, and 3623 differentially expressed genes at 33 °C, 10 °C, and 25 °C respectively, including 1197, 2735, and 1610 downregulated and 1589, 1921, 2013 genes upregulated at 33 °C, 10 °C, and 25 °C respectively (Appendix A). However, 795 differentially expressed genes were common at all temperatures (Appendix A). Gene Ontology (GO) and KEGG enrichment analysis shows that biosynthesis of secondary metabolites was downregulated in the mutant strains at all temperatures (Figure 6A–C), revealing that MoSch9 deletion affected the production of secondary metabolites.

Most of the differentially expressed genes were observed at low-temperature stress (10 °C). Consequently, the detailed study of KEGG enrichment analysis (Appendix A) and Gene Ontology (GO) for down and upregulated genes (Figure 6) in ΔMoSch9 strains at 10 °C revealed many enriched GO terms for downregulated genes including those involved in carbohydrates and polysaccharides metabolic process, cellular homeostasis, part of cell wall and membrane, and multiple coenzymes, chitin, and iron ion binding (Figure 6D). In contrast, the upregulated genes include those that are involved in carbohydrate derivative metabolism, glycosylation, part of Golgi apparatus, GTPase and transmembrane transport activity, etc. (Figure 6E). In our dataset, we also found some highly downregulated genes in relation to low temperature stress including Acetyl transferase1 (MGG_05861), mitochondrial thiol peroxidase (MGG_08256), Endoglucanase 3 (MGG_10083) and 4 (MGG_07575), ATP hydrolyzing-AMP binding (MGG_15681), Transaldolase (MGG_02624), Superoxide dismutase (MGG_02625), ABC transporter protein (MGG_11025), HSP70 (MGG_09631), LysM (MGG_10097), Semi aldehyde dehydrogenase (MGG_00385), Cytochrome p540 (MGG_08498) (Figure 6F) and many others including peroxidase, GAPDH, Transaldolase, etc. These results indicate that during cold stress, MoSch9 maintains cell wall and membrane integrity and also regulates the metabolic pathways, cold tolerance genes, and CYP 540 for secondary metabolite synthesis to maintain cellular homeostasis, important for fungal survival under cold stress conditions, as shown in the hypothetical model of the MoSch9 molecular mechanism during cold stress (Figure 7).

## 4. Discussion

In fungi, protein kinase cAMP (PKA) and TOR pathways are the major signaling pathways [54,55,56] that have been reported in yeast to work in parallel for cell physiological development and nutrient sensing [57]. Serine/threonine Mitogen-activated protein kinases (MAPKs) are the key enzymes of these phosphorylation cascade systems that play an important role in the regulation of various developmental processes as well as the transmission of diverse extracellular signals [55,58,59]. A commonly found eukaryotic serine/threonine protein kinase Sch9 belonging to an AGC kinase family has been reported to play an essential role in many of the cell’s physiological and signaling pathways, including growth, reproduction, pathogenesis, and stress tolerance [31,32,33,34,35,60,61,62]. In previous studies of Sch9 in different fungi, the protein has been involved in both cAMP (PKA) and TOR pathways in response to nutrient sensing, in virulence, and in autophagy [34,55,63].

Although Sch9 orthologs are well conserved in many pathogenic fungi, especially in ascomycetes filamentous fungi [31,32,47,61,62,64,65,66], they are functionally characterized in only few plant pathogenic fungi [31,66]. Our study showed that all the fungal Sch9 proteins characterized so far share conserved amino acid sequences at the ATP binding site, and that the active site reveals that Sch9 is conserved in fungi. We also demonstrated that MoSch9 shared closer phylogenetic linage with soil-borne entomopathogenic fungi, i.e., *B. Bassiana,* and hemibiotrophic fungi such as *F. graminarium*. Both these pathogens’ infection route results in the killing of their host [67,68], which is in line with our results that MoSch9 likely contributes to the pathogenic development of filamentous fungus during the necrotrophic phase. This was further confirmed when the MoSch9 expression pattern was observed to be upregulated during the stalk formation stage and at the necrotrophic phase of the pathogen disease cycle.

Asexual sporulation (conidiation) is the most common way of reproduction for this notorious fungus which involves the formation of mitotically derived non-motile spores called conidia that result from the “blowing-out” of conidiophores [48]. The Sch9 gene has already been reported to regulate the sporulation in pathogenic fungi [31,33,47]. In our study, similar to results in *C. albicans* [47], ΔMoSch9 mutants showed no influence on the growth rate of rice blast fungus that was previously reported to be affected in various fungi [31,33,60,61,66] due to the loss of Sch9. However, there was a significant reduction in conidiophore stalk production, spore numbers, germination, and appressorium formation, similar to the Sch9 deletion mutants of *A. fumigatus* [62], *F. graminarium* [31], *C. albicans* [47], and *C. higginsianum* [66], etc. This indicates the conserving role of Sch9 in the development of pathogenicity-related structures in filamentous fungi by affecting other conidiogenesis-related protein expression that helps the pathogen in the progression of its infection cycle [69,70,71,72,73]. In our study, we also observed that due to the delay in germination and appressorium formation in MoSch9 mutant strains, the pathogenicity of the respective strain was significantly affected. Similar results were observed when Sch9 was deleted in other pathogenic fungi [31,47,60,61,62,66].

In addition, we observed the role of Sch9 in hyphal-mediated pathogenesis that has not been reported before. Interestingly, in our study, MoSch9 deleted mutants completely lost the hyphal-mediated pathogenesis due to their inability to form hyphal tip appressorium-like structures. Similar results were obtained for the root infection assay, which further confirms our hypothesis of MoSch9 involvement in off-season survival: as previously reported, *M. oryzae* can survive in the form of hyphae within infected crop tissues [22,24], and loss of MoSch9 not only attenuated the hyphal infection but also restricted its growth during various stress conditions, i.e., cell wall, oxidative and osmotic stress that has also been reported in other fungi [31,47,60,61,62,66]. This indicates a conserving role of Sch9 during environmental stress. However, no study has yet been reported on its role in overwintering stress tolerance.

As fungi are present in all kinds of habitats, they have evolved elaborate strategies and can survive under adverse environmental conditions they may encounter during their life cycle, including the cold habitats that they occupy on almost 2/3rd of the land area on earth [74]. In fact, many fungal pathogens have been reported to overwinter in the form of different resting structures in a host population during off-season [6,7,12,19,22,24]. Rice blast fungus, *Magnaporthe oryzae*, has also been reported to overwinter; however, it forms hardly any resting structure. Instead, the pathogen undergoes various genetic modifications for its off-season survival in the form of conidia or mycelia to spread the infection in the following year [22,23,24]. The pathogen can increase its freezing tolerance in response to overwintering stress via multiple signal transduction pathways that rapidly recruit transcription factors to alter stress-responsive gene expression that results in alleviating the stress [75]. Hence, when studying the role of kinase protein MoSch9 in *M. oryzae* cold acclimation capability, we found that the deletion of MoSch9 not only attenuated the growth of mutant strains at low temperature, but also reduced the hyphae and spore survivability after cold (−20 °C) treatment, thereby showing the regulatory role of Mosch9 in cold stress tolerance. More than a 2-fold increase in the expression pattern of MoSch9 in wild type and significant downregulation in the expression of other cold tolerance proteins [44] in mutant strains during low-temperature stress further confirms the key role of MoSch9 in cold acclimation capability of *M. oryzae*.

To gain an insight into the changes in the protein expression that may serve as key biomarkers of temperature stress [76], comparative analyses of wild-type and MoSch9 mutant proteomic profile were performed. In our study, *M. oryzae* proteomic profiles showed an increase in total number of spots at low-temperature stress compared to high and normal, similar to that observed in rock-inhabiting fungi [77]. Furthermore, KEGG enrichment analysis shows the deletion of MoSch9 repressed the production of secondary metabolites that are the key factors in the long-term survivability of any living organism and act as a signaling molecule during various stress conditions [78,79,80]. Similarly, the KEGG enrichment analysis and Gene Ontology (GO) in ΔMoSch9 strains at 10 °C revealed that a large number of proteins involved in energy metabolism pathways, in ABC transport, in part of cell wall and membrane, and cellular homeostasis were downregulatedMetabolic pathways are one of the main components that are crucial for energy regulation of the cell and hence for cell survival [81], and MoSch9 deletion resulted in the disruption of carbohydrate metabolic pathways, which is in accordance with the mutant’s compromised growth during cold stress. In addition, fungal ABC transporters have been previously reported to work as an efflux pump to move natural toxins or cytotoxic secondary metabolites [34,82]. In plants, ABC transporters helped the plant to survive in a cold environment [83]. As *M. oryzae* overwinters in plant debris, ABC transporter proteins may be involved in the fight against cold stress. Similarly, the fungal cell wall and membrane structure is a vital component that maintains cellular integrity and viability and cushions the cell from mechanical and osmotic stress [42,84]. Cold stress reduces integrity of plasma membrane membranes which leads to the leakage of intracellular solutes [84,85].

Furthermore, some highly downregulated proteins in relation to low-temperature stress identified in our study are the heat shock proteins [75,86,87,88], Cytochrome p540 [89], ATP hydrolyzing proteins [90,91], Semi aldehyde dehydrogenase, endoglucanase, and ABC proteins [83,92,93] which have already been reported to be an important factor during cold stress in maintaining cellular homeostasis and cell membrane integrity. From these results, we deduce that during cold stress, the fungal cell membrane integrity was compromised, and the deletion of MoSch9 affected the expression of ATP binding cassettes (ABC) and ATP hydrolyzing proteins [94], thus affecting the signaling pathways involved in the regulation of other cellular processes including metabolic pathways and secondary metabolite synthesis—both important for the survival of the fungus under stress conditions [79,89].

## 5. Conclusions

In summary, functional characterization and transcriptomic study of MoSch9 mutant strains revealed that the respective proteins contribute positively to asexual sporulation and all types of pathogenesis of the rice blast fungus. In addition, our study has provided an insight into the significance of MoSch9 as a possible molecular factor responsible for overwintering stress tolerance of filamentous fungi by regulating the biosynthesis of cell wall integrity, stress, and cold-responsive proteins required for the cellular homeostasis of *Magnaporthe oryzae*. From these results, we propose that MoSch9 plays an important role in the long-term survival of *M. oryzae* during off-season. These findings underscore the need to comprehensively examine the extent to which MoSch9 and its interacting proteins can affect the pathogen’s ability to withstand the overwintering stress at field level. This will be helpful in the future to generate sustainable anti-inhibitory Sch9 compounds that can be sprayed in the crop fields before the cultivation of the rice to avoid the blast outbreak next season.

## Figures and Tables

**Figure 1 jof-08-00810-f001:**
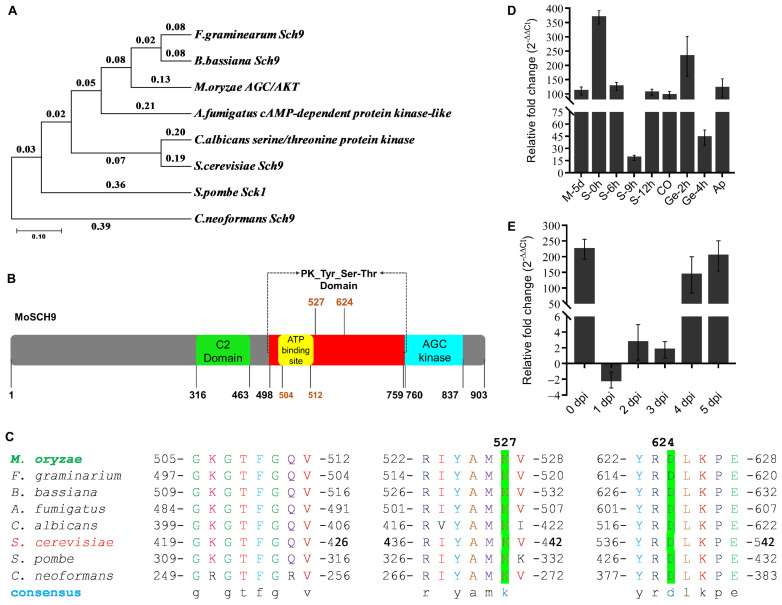
Phylogenetic assessment, protein structure, domain alignment, and expression of AGC/AKT protein kinase MoSch9 in *Magnaporthe oryzae*. (**A**) Maximum likelihood phylogeny constructed with Sch9 proteins homologs across different fungal species including *Fusarium graminarium*, *Beauveria bassiana*, *Magnaporthe oryzae*, *Aspergillus fumigatus*, *Candida albicans*, *Saccharomyces cerevisiae*, *Saccharomyces pombe*, and *Cryptococcus neoformans*. (**B**) MoSch9 protein structure. MoSch9 is a serine/threonine protein kinase. (**C**) Sequence alignment showing highly conserved ATP binding and active site in reported fungi. (**D**) Expression of MoSch9 at different growth phases of *M. oryzae*. For M, RBM; S, conidiophores stalks formation; Co, conidia; Ge, germination; Ap, appressoria. Vegetative hyphae on 5 day old CM were used as a control stage and assumed as unity (expression level of MoSch9 at hyphal stage on CM = 1). (**E**) Expression pattern of MoSch9 at different stages of infection development of *M. oryzae* in rice. Error bars indicate the means ± SD calculated from three independent technical replicates in which triplicate biological samples for each strain were examined in each experiment.

**Figure 2 jof-08-00810-f002:**
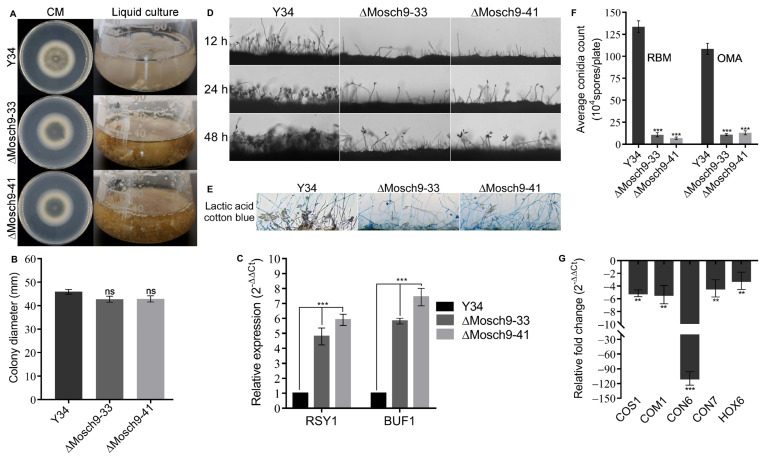
Vegetative growth and conidiogenesis assay. (**A**) Depicts the average colony growth and liquid culture of ΔMosch9 mutants compared to wild type on CM media for 7 days. (**B**) Statistical demonstration of the average growth rate of respective wild-type and mutant strains on CM. (ns) represents non-significant difference > 0.05. (**C**) Expression profiling of melanization-responsible genes in respective strain hyphae using qRT-PCR. (**D**) Represents ΔMosch9 mutants and wild type conidiophore development and their spore-bearing capabilities on rice bran medium. Bar = 10 μm. (**E**) Represents Lactophenol cotton blue-stained conidiophore stalks at 48 h. Bar = 20 μm. (**F**) Statistical data presenting conidiation capability of MoSch9 mutant strains relative to the wild-type strain. (**G**) qRT-PCR analysis of conidiation-responsible genes in MoSch9 mutant strains, taking wild type expression as control (unity = 1). Error bars indicate standard deviation from three independent replicates. (**) and (***) represent significant difference (*p* < 0.01 and *p* < 0.001, respectively) using *t* test.

**Figure 3 jof-08-00810-f003:**
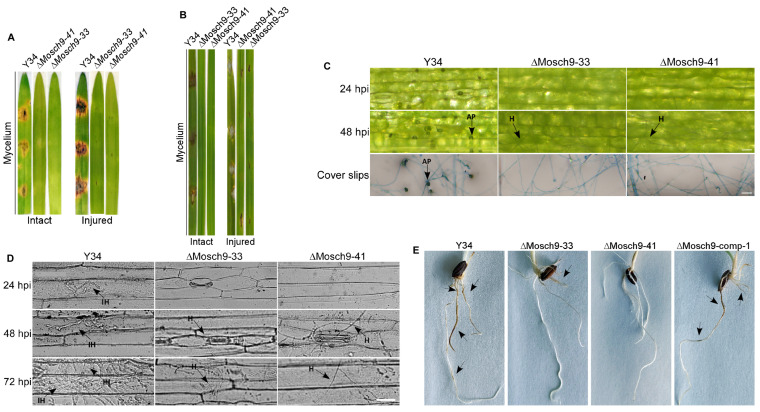
Impact of MoSch9 gene deletion on hyphal mediated pathogenicity, appressorium morphogenesis, penetration, and root infection capability of *M. oryzae*. Hyphal-mediated infection capability of the ΔMosch9 strains on intact and injured detached (**A**) barley and (**B**) rice leaves compared to wild-type strains. (**C**) Development of appressorium-like structures and (**D**) Hyphal mediated penetration on barley leaves inoculated with MoSch9 mutant strains and wild type mycelia. Scale = 20 μm. (**E**) Rice root infection evaluation at 5 dpi. Arrows show the infection sites. Nearly all results were obtained from three replicates.

**Figure 4 jof-08-00810-f004:**
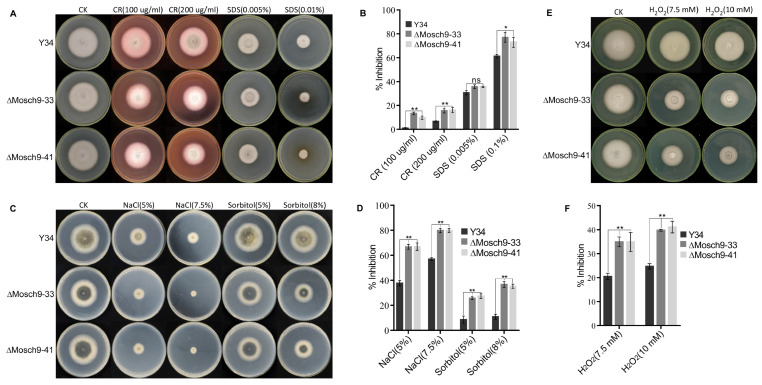
Contributions of MoSch9 to multiple environmental stress tolerance of *Magnaporthe oryzae*. (**A**) Depicts the average colony growth of ΔMoSch9 strains and wild-type cultured on CM media supplemented with different concentrations of cell membrane stress-inducing osmolytes (CR and SDS) for 7 days. (**B**) Statistical depiction of the growth inhibition rate for ΔMoSch9 strains, and wild type cultured on CR- and SDS-supplemented CM media for 7 days. (**C**) Showed the comparative colony growth of ΔMoSch9 strains and wild-type cultured on CM media supplemented with different concentrations of osmotic stress-inducing chemicals (NaCl and sorbitol) for 7 days. (**D**) Statistical illustration of ΔMoSch9 strains and wild type growth inhibition rate under NaCl and sorbitol stress. (**E**) Depicts the average hyphal growth of respective strains cultured on CM media supplemented with different concentrations of oxidative stress-inducing osmolyte, H2O2, for 7 days. (**F**) Statistical depiction of the growth inhibition rate for ΔMoSch9 strains and wild type under hydrogen peroxide (H2O2) stress. Inhibition rate was obtained using the formula: (Inhibition rate = (untreated strain diameter − treated strain diameter)/(diameter of untreated strain × 100%)). Error bars indicate the means ± SD calculated from three independent technical replicates in which triplicate biological samples for each strain were examined in each experiment. (ns), (*), and (**) represent significant difference with *p* > 0.05, *p* < 0.05 and *p* < 0.01, respectively using *t* test.

**Figure 5 jof-08-00810-f005:**
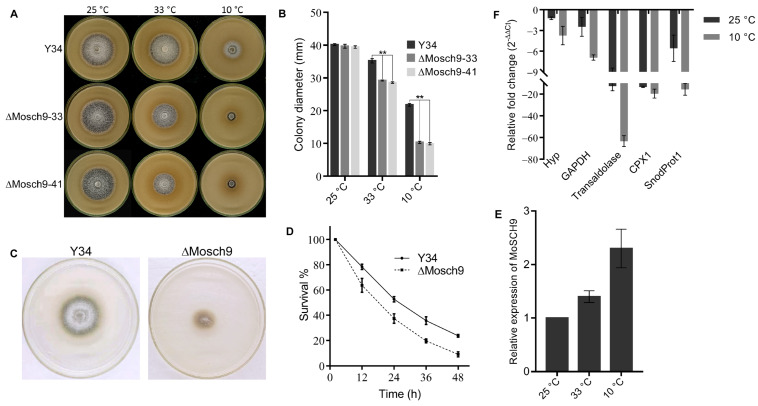
MoSch9 deletion affected the full morphological development of rice blast fungus under temperature stress. (**A**) Depicts the average colony growth of ΔMoSch9 strains and wild type cultured on RBM media under high (33 °C) and low temperature (10 °C) stress for 10 days, taking 25 °C as normal temperature. (**B**) Average growth rate of respective wild-type and mutant strains on RBM after 10 days of temperature stress. (**C**) Respective wild-type and mutant strain growth on CM medium for 5 days after −20 °C treatment. (**D**) Percentage survival rates of wild type and MoSch9 mutant’s conidia after −20 °C treatment. (**E**) Expression pattern of MoSch9 in *M. oryzae* under high (33 °C) and low temperature (10 °C) stress, taking *M. oryzae* growth at 25 °C as control. (**F**) qRT-PCR analysis of *M. oryzae* cold responsive genes i.e., Hyp (Putative uncharacterized protein), GAPDH (Glyceraldehyde-3-phosphate dehydrogenase), Transaldolase, CPX1 (Catalase-peroxidase 1), and SnodProt1 in MoSch9 mutant strains at normal (25 °C) and low (10 °C) temperature taking wild type expression as control (unity = 1). Error bars indicate standard deviation from three independent replicates. (**) represent significant difference (*p* < 0.01).

**Figure 6 jof-08-00810-f006:**
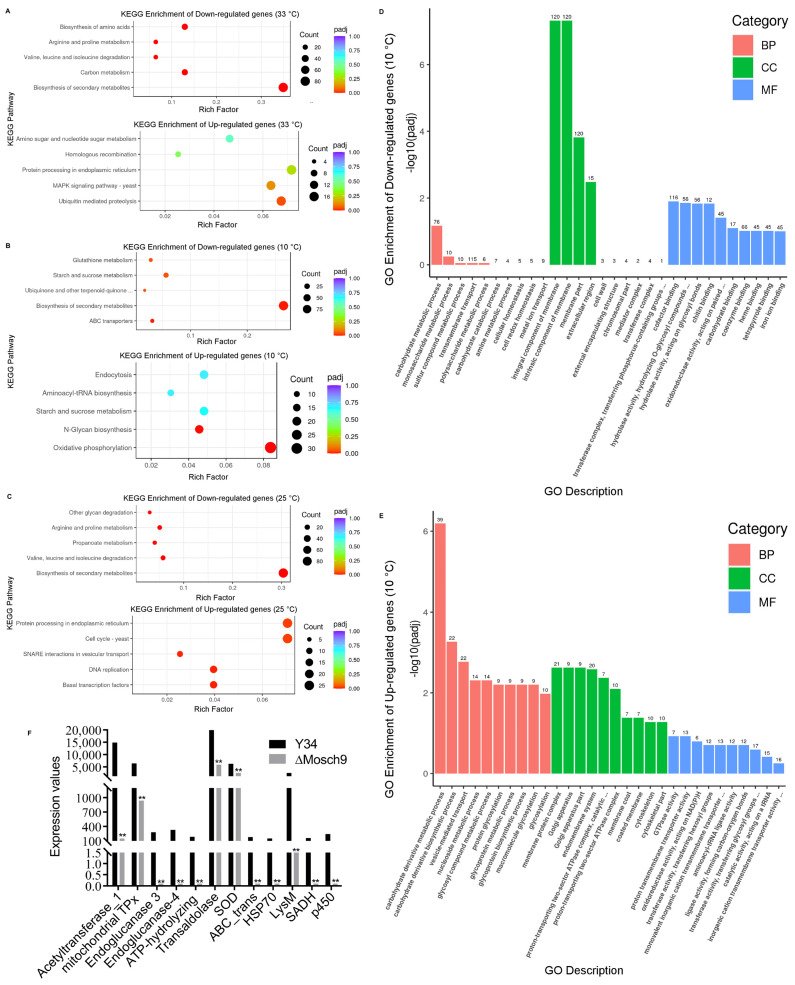
KEGG enrichment analysis of differentially expressed genes at various temperature stresses. KEGG enrichment analysis of downregulated and upregulated genes after ΔMoSch9 deletion at (**A**) 33 °C, (**B**) 10 °C, and (**C**) 25 °C. Gene Ontology (GO) based molecular functions of the genes (**D**) downregulated and (**E**) upregulated in MoSch9 mutants at a two-fold expression threshold. (**F**) Expression pattern of fungal genes repressed in MoSch9 mutants compared to wild type when subjected to 10 °C treatment. (**) represent significant difference (*p* < 0.01).

**Figure 7 jof-08-00810-f007:**
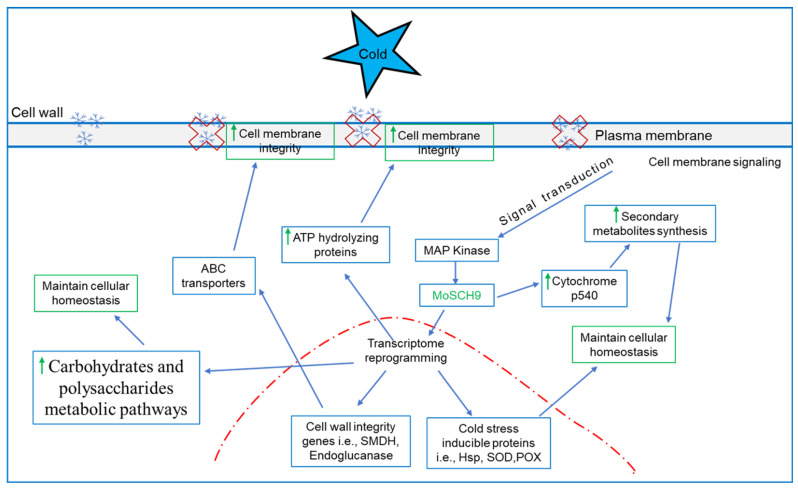
Hypothetical model of MoSch9 molecular mechanism during cold stress. During cold stress fungal cells use the cell wall and membrane protein signaling system to initiate MoSch9 expression in MAP kinase cascade system. which in turn initiates the gene reprogramming of ATP hydrolyzing proteins and cell wall integrity proteins to maintain cell wall and membrane integrity. In addition, MoSch9 regulates metabolic pathways, cold tolerance genes, and CYP 540 for secondary metabolite synthesis to maintain cellular homeostasis, important for the fungal survival under cold stress conditions.

## Data Availability

Not applicable.

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
