# Peer review of "AGC/AKT Protein Kinase SCH9 Is Critical to Pathogenic Development and Overwintering Survival in Magnaporthe oryzae"

_jof, 2022, doi:10.3390/jof8080810_

Round 1
Reviewer 1 Report
The manuscript (AGC/AKT protein kinase SCH9 is critical to pathogenic development and overwintering survival in Magnaporthe oryzae) is interesting as it describes the role of MoSch9 gene in the pathogenicity of rice blast pathogen M. oryzae.
The abstract
Please add some results to the abstract, be more specific and show the value of your study results. Add clear aims of the study.
Introduction
L65 The authors wrote (Similarly, rice blast pathogen Magnaporthe oryzae, now emerged as a wheat blast pathogen, has also been reported to overwinter through its long-term survival on seeds). This sentence is not accurate as both Magnaporthe oryzae pathotype Oryza in rice and Magnaporthe oryzae pathotype Triticum in wheat are genetically different and not the same pathogen. Please revise the recent papers in this respect and explain.
The end of the introduction looks like a conclusion. However, you should only state the objectives of the study without explaining results and conclusion.
Materials and methods: several details are missing for example the methods for targeted gene deletion of MoSch9 mutants, you should explain in brief how did you prepare the mutants.
Results
It is not good to include and discuss others work in the results section. Please transfer to the discussion section.
The conclusion is very long and includes some unsuitable sentences such as (Hypothetical model of MoSch9 molecular mechanism during cold stress in Fig 7 shows that during low temperature stress fungal cell use cell wall and membrane protein signaling system…..), it should be in results section using this format. Please summarize the conclusion highlighting only the main findings and the contribution in the field.
The references: all scientific names should be in italic such as L664, L675, L677, L684 and many others. The references should be carefully checked for wrong scientific names and format.
The manuscript should be revised carefully for English language.
Please do the following suggested changes in the manuscript.
L10 Change alteration to alterations
L14 Change to the molecular
L17 Change Play to Plays
L19 Change to the functional
L21 Change to long term
L33 Change to produce
L43 Change to the bacterial pathogen
L72 Change to survive on plants debris
L83 Grammatical error (has been should be have been)
L89 Change to affect fungal filamentation
L95 The authors wrote (After every two days, three blocks…) rephrase please.
L137 Grammatical error (was change to were carried out
L565 Change to is able to survive
L590 Change to gain an insight of changes
Author Response
We are grateful for the reviewer's positive feedback, thoughtful comments, and efforts towards improving our manuscript. We have addressed all the concerns of reviewer in our revised manuscript, that were mentioned in comment section. Furthermore, we have responded to each comment point-by-point in word file attached below.

Reviewer 2 Report
The subject of the manuscript is consistent with the scope of the Journal. The topic of research is interesting. The present paper is prepared in the usual manner for scientific work, both the division into chapters collected results in the form of figures and tables. The organization of the manuscript is satisfactory, and the length of manuscript is appropriate to the content.
I agree with the authors that research of this type is important.
Some comments
1. Figures 1-7 should be included in Chapter 3. "Results", section "3.1. Results description", and not be a separate section "3.2. Figures".
2. The description of the X and Y axes in Fig. 6 is illegible. The font is too small, please correct them.
3. Please, be sure that all the references cited in the manuscript are also included in the reference list and vice versa with matching spellings and dates.
Author Response
We would like to thanks and appreciate the reviewer's positive and constructive feedback towards improving our manuscript. We have addressed all the comments of reviewer in our revised manuscript. Furthermore, we have responded to each comment point-by-point in word file attached below.

Round 2
Reviewer 1 Report
The manuscript (AGC/AKT protein kinase SCH9 is critical to pathogenic development and overwintering survival in Magnaporthe oryzae).
Thank you for your answers and corrections.
L209 Please add the conditions of TRIzol reagent kit.
Please add details about cDNA synthesis and PCR conditions.
Still I can find grammatical errors in the manuscript such as:
L12 Change to relies on the physiological alteration
L21 Change to most importantly, the functional
L134 Change to fragment within...
L656 Change to all types of pathogenesis.
L660 Please keep the following format (From these results, ....).
Please check the manuscript carefully for English errors.
Author Response
We would like to thanks and appreciate the reviewer's positive and constructive feedback towards improving our manuscript. We have improved our manuscript in material methodology section that can be seen on page 208-219 in revised manuscript. Also we have gone through manuscript twice to correct the grammatical errors.